# Internal calibration for opportunistic computed tomography muscle density analysis

Ainsley C. J. Smith[1,2,3], Justin J. Tse[2,3], Tadiwa H. Waungana[1,2,3], Kirsten N. Bott[2,3], Michael T. Kuczynski[1,2,3], Andrew S. Michalski[1,2,3], Steven K. Boyd[1,2,3], Sarah L. Manske[1,2,3]*

1 Biomedical Engineering Graduate Program, University of Calgary, Alberta, Canada, 2 Department of Radiology, Cumming School of Medicine, University of Calgary, Alberta, Canada, 3 McCaig Institute for Bone and Joint Health, University of Calgary, Alberta, Canada

* smanske@ucalgary.ca

## Abstract

### Introduction

Muscle weakness can lead to reduced physical function and quality of life. Computed tomography (CT) can be used to assess muscle health through measures of muscle cross-sectional area and density loss associated with fat infiltration. However, there are limited opportunities to measure muscle density in clinically acquired CT scans because a density calibration phantom, allowing for the conversion of CT Hounsfield units into density, is typically not included within the field-of-view. For bone density analysis, internal density calibration methods use regions of interest within the scan field-of-view to derive the relationship between Hounsfield units and bone density, but these methods have yet to be adapted for muscle density analysis. The objective of this study was to design and validate a CT internal calibration method for muscle density analysis.

### Methodology

We CT scanned 10 bovine muscle samples using two scan protocols and five scan positions within the scanner bore. The scans were calibrated using internal calibration and a reference phantom. We tested combinations of internal calibration regions of interest (e.g., air, blood, bone, muscle, adipose).

### Results

We found that the internal calibration method using two regions of interest, air and adipose or blood, yielded accurate muscle density values (< 1% error) when compared with the reference phantom. The muscle density values derived from the internal and reference phantom calibration methods were highly correlated ($R^2 > 0.99$). The coefficient of variation for muscle density across two scan protocols and five scan positions was significantly lower for internal calibration (mean = 0.33%) than for Hounsfield units (mean = 6.52%). There was no

**Data Availability Statement:** All relevant data are within the paper and its Supporting Information files.

**Funding:** ACJS and MTK are supported by Natural Sciences and Engineering Research Council (NSERC, Canada) fellowships. JJT is supported by the Canadian Institutes of Health Research (CIHR) Postdoctoral Fellowship. Study funding was provided by NSERC Discovery Grants (RGPIN-2019-04135; RGPIN-2018-03908) and the American Society for Bone and Mineral Research Rising Star Award (SLM). The funders had no role in study design, data collection and analysis, decision to publish, or preparation of the manuscript.

**Competing interests:** The authors have declared that no competing interests exist.

difference between coefficient of variation for the internal calibration and reference phantom methods.

## Conclusions

We have developed an internal calibration method to produce accurate and reliable muscle density measures from opportunistic computed tomography images without the need for calibration phantoms.

## Introduction

Muscle weakness is associated with physical impairment, reduced quality of life, and mortality [1,2]. Computed tomography (CT) is a widely used clinical imaging modality that can be utilized to evaluate muscle degeneration through quantitative measures of muscle cross-sectional area and density. Muscle cross-sectional area is an indicator of muscle health that is correlated with measures of muscle function (e.g., handgrip strength, knee extension strength, short physical performance battery scores) and can be used to predict health outcomes (e.g., frailty, length of hospital stay) [3–6]. However, structural alterations, such as fat infiltration, may result in muscle cross-sectional area values underestimating changes in muscle function and muscle strength. Fat content within muscle structures can be evaluated through measurements of muscle density since fat infiltration leads to lower density muscle [7–11]. Lower muscle density in the lumbar region and the lower limbs is correlated with reduced muscle strength [3,12,13], and can be combined with measures of muscle cross-sectional area for a more comprehensive assessment of muscle health.

Muscle cross-sectional area and density can be assessed through CT imaging. Chest and abdominal CT images are frequently acquired in hospital settings for clinical diagnoses and these clinically acquired CT scans have been repurposed to investigate muscle cross-sectional area, and muscle attenuation as a surrogate for density, at the level of the lumbar vertebrae without exposure to additional ionizing radiation [14–17]. While density can be inferred from the X-ray attenuation or grey-scale value in the CT images, natively represented in Hounsfield units (HU), these values are dependent on the X-ray energy, photon flux, object position within the scanner bore, spatial resolution, beam hardening, and other artifacts [11,18–23]. This makes it challenging to reliably compare HU as a proxy for muscle density between different scanners and scan protocols. A standard solution for bone mineral density (BMD) assessment is to calibrate HU to bone analogous material density values using phantoms made with materials of known dipotassium phosphate or calcium hydroxyapatite (HA) densities [24,25]. Previous research has used BMD phantoms to successfully quantify muscle and lipid composition with a semi-automatic segmentation method based on histograms and anatomical location [23,26]. However, the use of BMD phantoms yield mineral density values in mg of dipotassium phosphate or calcium hydroxyapatite per cm$^3$ [27], whereas muscle is largely composed of water. Muscle density values in g/cm$^3$ of water could be more intuitively interpreted. However, water-based reference phantoms are not widely available due to challenges in mimicking soft tissues with shelf-s\ materials and absence of consensus of appropriate materials [28].

As an alternative to phantoms, internal density calibration techniques have been developed to convert HU to BMD values [29–32]. These methods have been developed for opportunistic BMD analysis because phantoms are not typically included in clinical CT scans or used for

clinical diagnoses. Internal density calibration involves selecting regions of interest (ROIs) of tissues (e.g., bone, blood, muscle, adipose, air) as reference materials. Some methods calibrate HU values of these ROIs to previously determined tissue-equivalent reference values [29–31]. However, a potentially more robust method involves estimating the effective scan energy using mass attenuation values of materials within the scan and then deriving the relationship between HU and material density [32]. Unfortunately, it is unknown whether BMD internal calibration methods are applicable to muscle density analysis since muscle attenuates X-rays to a lesser extent than bone. To date, there has yet to be an internal calibration method developed or validated for quantitative muscle density analysis.

The objective of this study was to design and validate an internal density calibration technique to directly measure muscle density from CT images. The basis of this internal calibration method relies upon a previously validated method for BMD [32]. Using bovine tissue samples, we sought to test the optimal combination of ROIs for muscle density analysis using the internal calibration method, as well as to compare the internal calibration method with the reference phantom calibration method. We hypothesized that our internal calibration method would be comparable to the reference phantom calibration method and would be a more robust measure of muscle density than native HU.

## Methodology

### Imaging

We used a standardized scan setup composed of muscle, cortical bone, adipose, blood, and a reference phantom, as shown in Fig 1. We scanned 10 bovine muscle samples which were acquired, stored, and scanned under the same conditions. The sample size was selected to

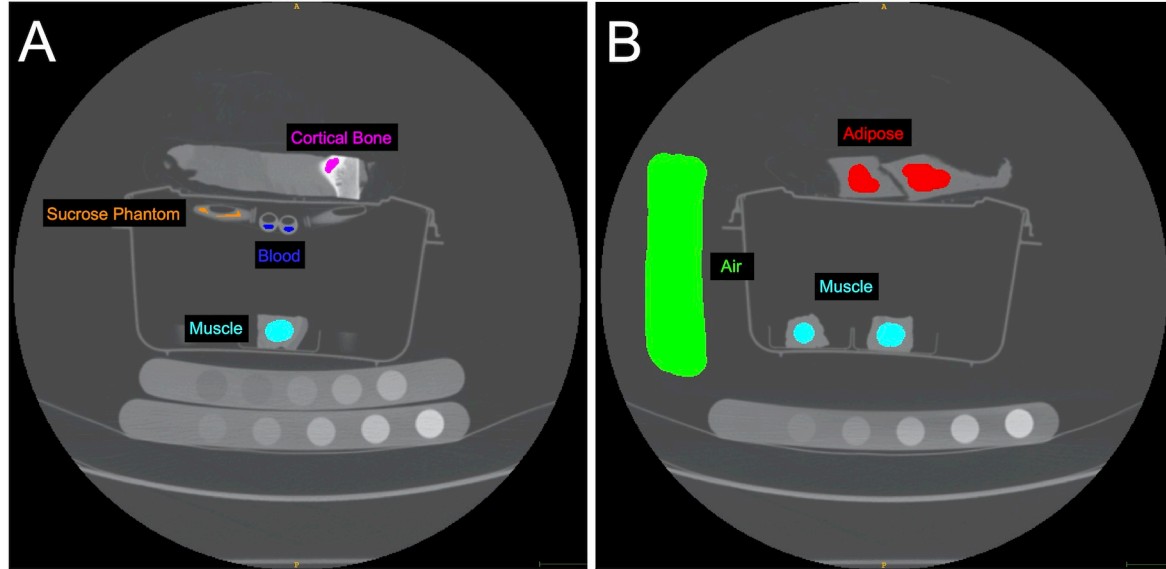

**Fig 1. Computed tomography (CT) image showing the reconstructed field of view (36 cm diameter).** Internal calibration regions of interest (ROIs) (air, muscle, adipose, bone, blood) were selected in the axial plane. The scan included various tissue samples, a custom sucrose water phantom, and two traditional bone phantoms (Bone Density Calibration Phantom BDC-6, QRM GmbH; Model 3 CT Calibration Phantom, Mindways Inc.). The (hydroxyapatite) HA bone phantom was used to validate the sucrose water phantom (S1 Fig). A) Axial slice including cortical bone, blood, one of the ten muscle samples, and the sucrose water phantom. B) Axial slice including adipose, air, and two of the ten muscle samples. Note that the ten muscle samples were all included in the scan, although they are not all visible in the slices shown.

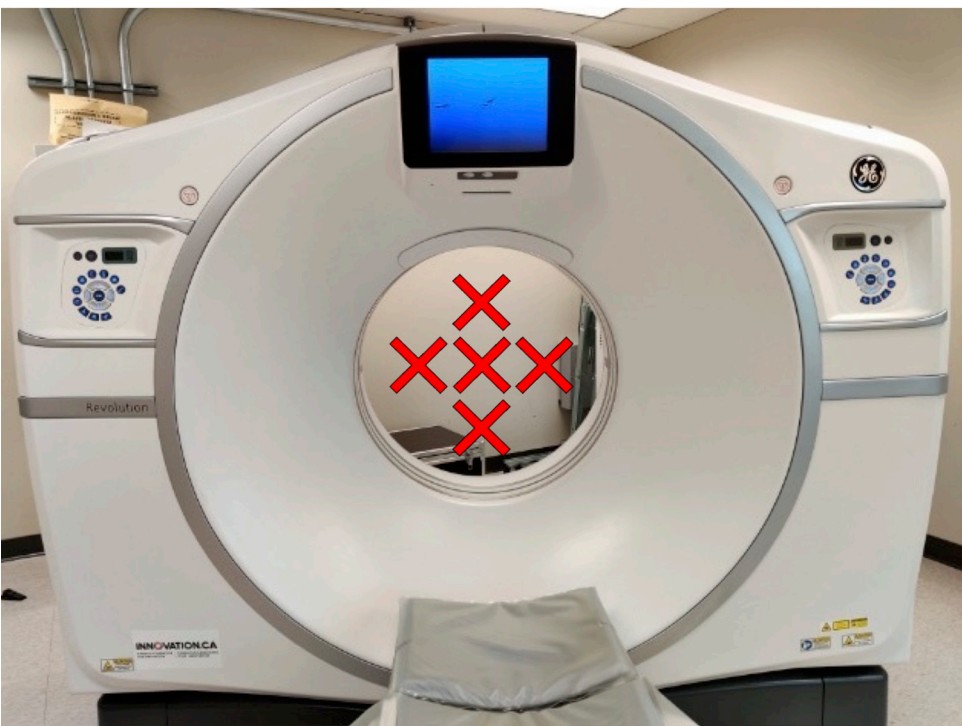

**Fig 2. Muscle sample positioning.** The muscle samples were scanned at five different positions within the scanner bore (70 cm diameter): left, right, top, bottom, and centre. The position of the samples was altered by adjusting the table positioning. Note that the scan setup was approximately 19 cm in height and could therefore be moved into different positions by adjusting table positioning.

provide a range of fat infiltration. Bovine bone, bovine adipose, and swine blood were included in these scans and used as ROIs for the internal calibration method. Swine blood was used instead of bovine blood as it is more readily available as a fresh sample. As muscle density phantoms are not widely available, we included a custom sucrose water phantom consisting of five vials of sucrose water concentrations with densities of 1.00g/ml, 1.01g/ml, 1.04g/ml, 1.06g/ ml, and 1.11g/ml. We scanned the muscle samples, ROI samples, and phantoms (sucrose water phantom; HA Bone Density Calibration Phantom BDC-6, QRM GmbH; Dipotassium Phosphate Model 3 CT Calibration Phantom, Mindways Inc.) using a clinical CT scanner (GE Revolution HD GSI, GE Healthcare, Chicago, USA). The HA bone phantom was used for comparison with BMD calibration [23]. We used two different scan protocols (abdominal kidney, urine, bladder (KUB) thins protocol and chest mediastinum protocol) and five different object positions (right, left, top, bottom, centre, as shown in Fig 2) to test the reliability of the internal calibration method across different scan conditions. These scan protocols were selected because they are commonly used in the clinical setting [33,34]. The KUB thins protocol was performed with 120 kVp, 99 mAs, 1.250 mm slice thickness, 1.37 mm spiral pitch factor, and was reconstructed with a standard convolution kernel. The chest mediastinum protocol was performed with 120 kVp, 90 mAs, 0.625 mm slice thickness, and 0.98 mm spiral pitch factor, and was reconstructed with a standard convolution kernel. All scans were acquired on the same day to reduce the effect of CT scanner fluctuation over time. The CT scanner undergoes quality control daily. Ethics approval was not required for this study as only commercial meat products were used and there was no contact with live animals.

## Calibration

We used the internal calibration approach previously described [32] (https://github.com/Bonelab/Ogo) as a basis and developed an approach specific for muscle. This approach relates the mass attenuation coefficients attained from the National Institute of Standards and Technology (www.nist.gov) and the measured HU of internal ROIs (blood, bone, air, fat, and muscle) to estimate the effective scan energy [32]. The relationship between HU and water equivalent density is then computed through an optimization process that estimates the effective scan energy by iteratively relating the mass attenuation coefficients and HU values for each selected ROI [32]. We optimized the internal calibration method for muscle density analysis by determining the effects of different combinations of material ROI selection on the output muscle density values compared with those derived from the reference sucrose water phantom calibration method. We selected ROIs for air, blood, muscle, adipose and cortical bone regions throughout the scan in the axial plane (Fig 1). We compared internal calibration based on 1) all material ROIs (air, adipose, blood, muscle, bone) as per the method described in Michalski et al. [32]; 2) air, adipose, blood, muscle; 3) air, adipose, blood; 4) air, blood; and 5) air, adipose. We added the output of a water-equivalent density image for muscle density analysis, as the original image processing pipeline only included a dipotassium phosphate-equivalent density image output. The 10 scans (2 protocols, 5 positions) were each individually calibrated using the internal calibration method based on the material ROI combinations described above, the reference sucrose water phantom, and the HA bone phantom. For the reference sucrose water phantom calibration method, observed HU values were rescaled with the linear equation derived from the calculated densities of the sucrose vials, based on the known concentration of sucrose. For the HA bone phantom calibration method, the linear equation was derived from the density of the HA rods. All calibration methods involved manually segmenting the bovine muscle samples in ITK-SNAP V3.8.0 [22], a medical image visualization and analysis software, and converting the observed HU into muscle density values (g/cm$^3$ or mgHA/cm$^3$). To accurately assess the reliability of the calibration methods across scan conditions, we used image registration (aligned the images over top of one another) to ensure that muscle density was measured at the same ROI location for all 10 scans. Image registration was performed using an initial alignment of images through landmark based rigid 3D transformation, followed by a final alignment based on optimizing mutual information between images [35]. All image analysis was conducted using ITK-SNAP V3.8.0 [22] and Python 3.6.10 and 3.8.5 with numpy 1.20.3 [36], VTK 9.0.1 [37], and SimpleITK 2.0.2 [38] libraries.

## Statistical analysis

Statistical analyses were performed using R (version 4.1.0) and RStudio (Version 1.4.1106). Linear regression was performed to compare muscle density values derived from the internal calibration method with those derived from the reference sucrose water phantom and the HA bone phantom method. Bland-Altman analysis was conducted to estimate the mean bias associated with the internal calibration method when compared with the reference sucrose water phantom method and to assess the performance of the internal calibration method when different ROI combinations were used. Linear regression was performed to assess any association between bias and the magnitude of density. Lastly, the coefficient of variance (CV) values for muscle density of each sample across the 10 scan conditions was calculated. The mean CV for the internal calibration method, the reference sucrose water phantom calibration method, and native HU were compared using within-subjects ANOVA and post-hoc analysis with Bonferroni. Statistical significance was set to $p < 0.05$.

## Results

The mean muscle density and standard deviation for the 10 samples was 1.0636 (±0.0086) g/cm$^3$ as derived from the internal calibration method (air and adipose ROIs) and 1.0694 (±0.0086) g/cm$^3$ as derived from the reference sucrose phantom calibration method, acquired from the chest scan positioned in the centre of the scanner bore. The mean HU and standard deviation for the 10 samples was 57.1077 (±8.5507), acquired from the chest scan positioned in the centre of the scanner bore. We found that the exclusion of bone as a material ROI in the internal calibration process dramatically reduced the error of the derived muscle density values when compared to the reference sucrose water phantom calibration derived muscle density values (bone included: mean error = -0.055, 95% Limits of Agreement (95%LOA) = 0.0020; bone excluded: mean error = -0.004, 95%LOA = 0.0001) (Fig 3). After excluding bone, the exclusion of muscle (mean error = -0.004, 95% LOA: 0.0001), and then blood (mean error = -0.006, 95% LOA: 0.0001) or adipose (mean error -0.004, 95% LOA: 0.0001) ROIs had a limited effect on the derived muscle density values. We completed the remainder of the analysis using two ROIs, adipose and air, which were easily identifiable and produced accurate muscle density values with the internal calibration method as described above.

We found that the muscle density values derived from the internal and reference sucrose water phantom calibration methods were highly correlated ($R^2 > 0.99$) (Fig 4). The internal muscle density calibration method slightly underestimated muscle density, but the error was low when compared to the reference sucrose water phantom method ($< 0.006$ g/cm$^3$) with only a 0.54% difference between the muscle density values derived from the two methods (Fig 3). We found that with increasing muscle density, the error associated with the internal calibration method diminished linearly (y-intercept = -0.01, slope = 0.006).

The mean CV for density of muscle samples across different scan protocols and positioning was significantly different across the internal calibration method, sucrose water phantom method, and native HU ($p < 0.001$) (Fig 5). The mean CV for density of the muscle samples was significantly greater the native HU (mean CV = 6.52%) when 1) compared to the internal muscle density calibration method (mean CV = 0.33%) ($p < 0.001$) and 2) compared to the

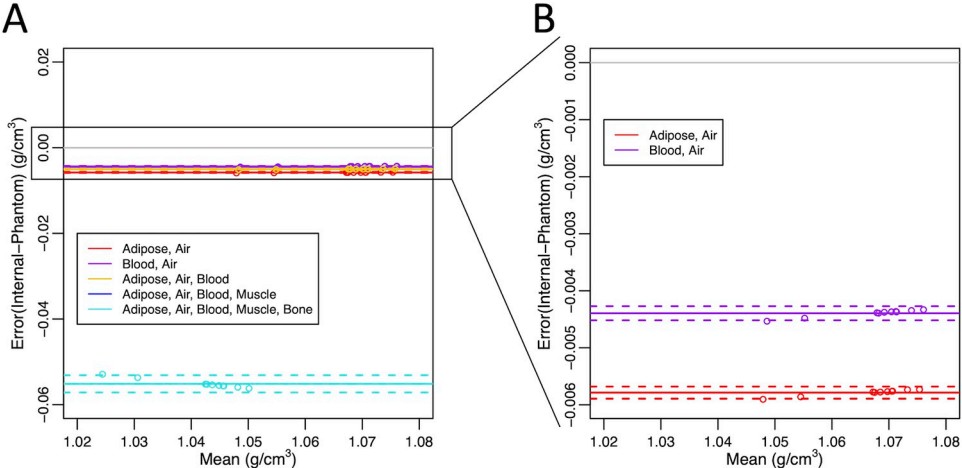

**Fig 3. Bland-Altman plots comparing calibration methods.** Bland-Altman plots comparing the differences between internal derived muscle density values and reference sucrose water phantom derived muscle density values (n = 10 muscle samples). Different combinations of regions of interest (ROIs) for the internal calibration method are represented in different colours. The mean differences are shown as solid lines and 95% limits of agreement are shown as dashed lines. A) Bland-Altman plot with all tested internal calibration ROI combinations. B) Bland-Altman plot with two internal calibration ROI combinations (air, adipose; air, blood).

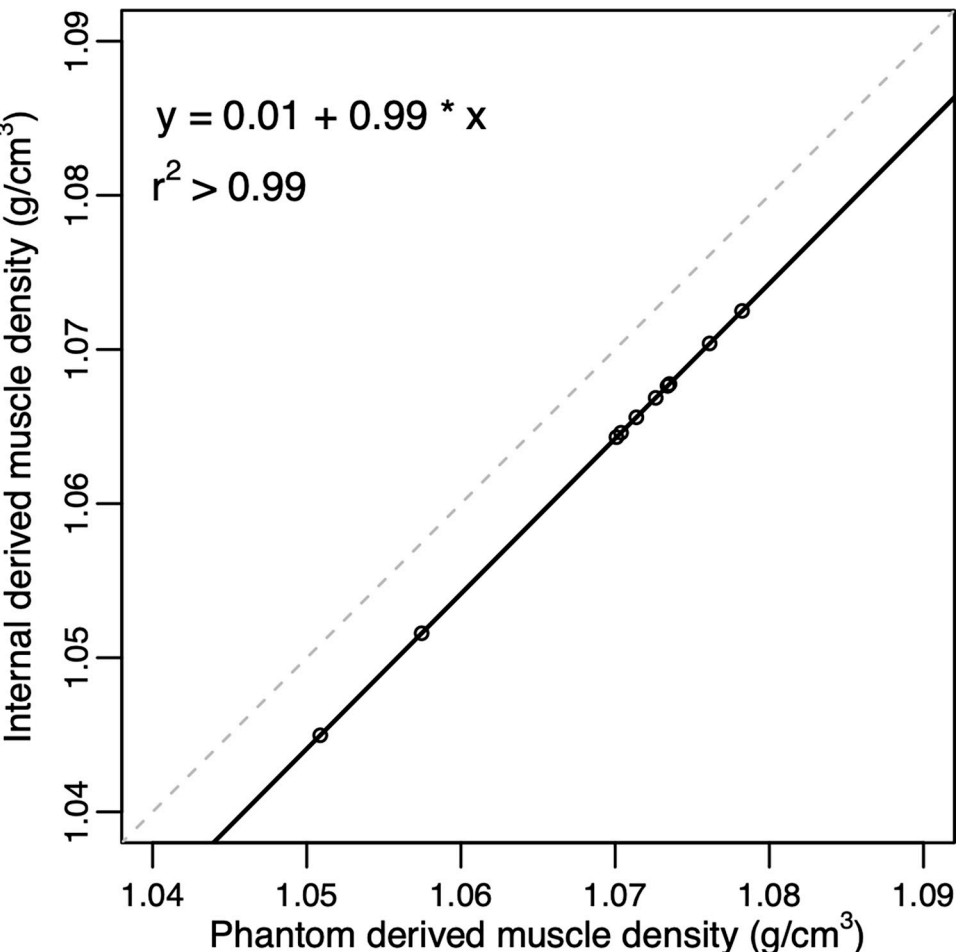

**Fig 4. Regression plot comparing calibration methods.** Regression plot comparing the muscle density values (n = 10 muscle samples) derived from the internal calibration method and the reference sucrose water phantom method. Solid line indicates regression line, dashed line indicates the line of unity.

reference sucrose water phantom calibration method (mean CV = 0.52%) (p < 0.001). There was no difference between the mean CV for density of the muscle samples derived from the internal calibration method and the reference sucrose water phantom calibration method (p = 1).

The HA bone phantom derived muscle density values were highly correlated with those derived from the reference sucrose water phantom calibration method ($R^2 > 0.99$) and the internal calibration method ($R^2 > 0.99$) (S1 Fig). However, the absolute values differed because the reference sucrose water phantom and internal calibration methods derive muscle density in g/cm$^3$, whereas the HA bone phantom derives muscle density values in mgHA/cm$^3$ (S2 Fig).

## Discussion

We implemented and validated a previously established CT internal calibration method that estimates scan effective energy for muscle density analysis [32]. The results support our hypothesis that the internal muscle density calibration method is comparable to the reference sucrose water phantom calibration method. We also found that, as hypothesized, the internal

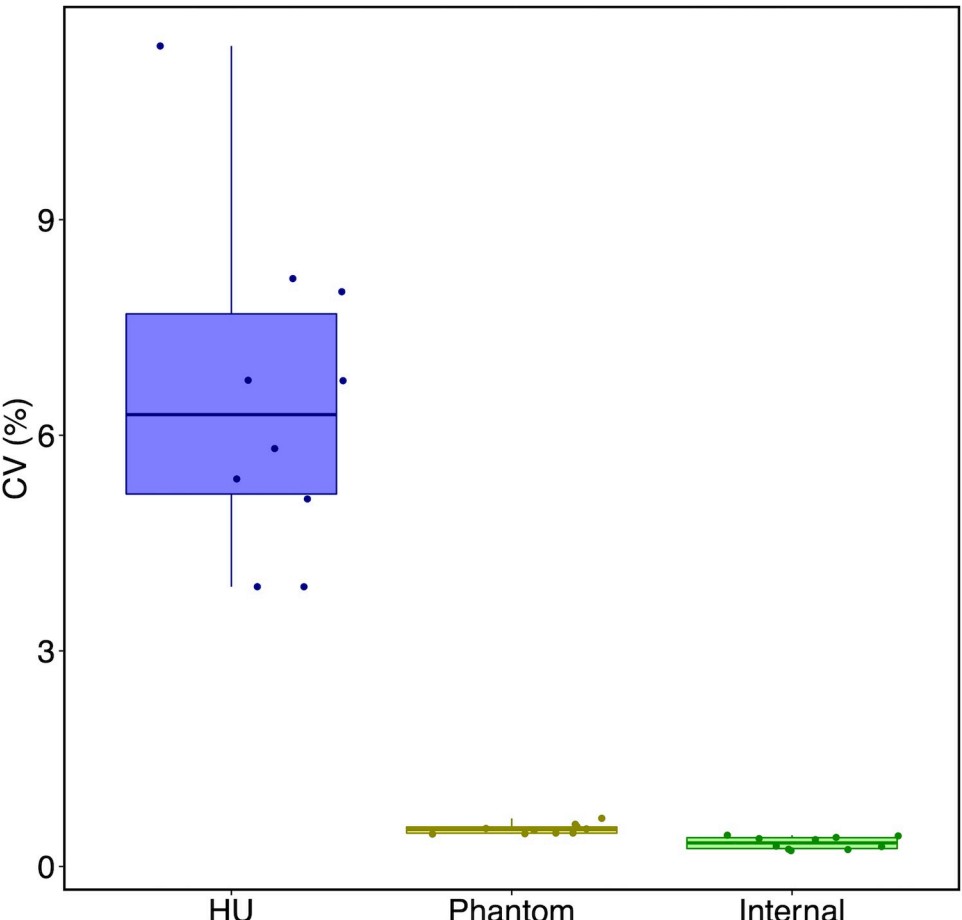

**Fig 5. Boxplot comparing coefficient of variation (CV) values.** Coefficient of variation (CV) for muscle density (n = 10 muscle samples) values across different two scan protocols and five scan positions. The left boxplot represents native Hounsfield units (HU). Muscle density values were derived from the reference sucrose water phantom calibration method (middle boxplot) and the internal calibration method (right boxplot). Each box extends from the 25th percentile to the 75th percentile of the distribution of the respective values. The horizontal lines within each box denote median values and the vertical lines outside the box denote the minimum and maximum values.

calibration derived muscle density values were more robust than HU as demonstrated across 5 scan positions and 2 scan protocols.

We demonstrated that using air and adipose ROIs for the internal calibration method were sufficient to yield accurate muscle density values when compared with the reference sucrose water phantom calibration method. When bone was excluded as an ROI, there was a large reduction in error associated with the internal calibration method. This error may have been associated with tissue inhomogeneity within the bone ROI, paired with the high density of bone relative to muscle. Inhomogeneity in cortical bone could affect the calculated slope of the linear equation and result in drastic differences in perceived muscle values. Besides bone, we excluded muscle as an internal calibration ROI to avoid bias in the output muscle density values. Blood was excluded as an ROI to simplify the methods and because the vessels are relatively small, which makes it challenging to select regions composed only of blood. However, blood could be used to substitute the adipose ROI if necessary. Our tests revealed that the removal of muscle and blood ROIs did not greatly impact our measured muscle density values. While optimal ROI selection may differ for BMD internal calibration, Lee et al. similarly

found that air and adipose (or blood) internal calibration ROIs produced precise BMD values [31]. This differs from other evaluations which used air, adipose, blood, skeletal muscle [39], and bone [32]. However, Bartenschlager et al. recently quantified accuracy error associated with different ROIs and ultimately suggested the use of air and subcutaneous adipose tissue or blood [40].

Consistent with BMD internal calibration studies, we found our internal muscle density calibration method resulted in muscle density values that were highly correlated with those produced by the phantom calibration method (in our study, the sucrose water phantom and the HA bone phantom) [31,32,41]. While our internal muscle density calibration method underestimated sucrose water phantom derived muscle density, the error was small ($< 1\%$) compared to clinical studies that have observed 10–40% differences in muscle attenuation between healthy controls and patients with hip osteoarthritis, fractures, or aortic disease [3,42,43]. We found that our internal calibration method was more accurate, when compared to the sucrose water phantom, with increasing density values. Hence, this method may be more accurate when applied to denser muscle. Nonetheless, our internal calibration method was significantly less sensitive to scan protocol and object positioning within the scanner bore when compared to native HU, which is how CT derived muscle density values are usually reported in the literature [7–9,14–17,44,45]. Similarly, other studies have noted inconsistencies in absolute HU values across scan protocols and positioning [11,18–23]. Therefore, rather than reporting native HU, we recommend an internal calibration procedure to convert to g/cm$^3$.

There are important limitations associated with our study. Notably, ex vivo bovine and swine tissue samples were scanned instead of human participants, which may have different tissue densities that influenced the internal calibration derived muscle density values. These samples do not account for differences in patient anatomy (i.e., patient size) which have been noted to impact resulting HU values [46–48]. However, the internal calibration approach utilized here should be less dependent on patient characteristics as reference values are taken from the patient themselves, which should account for patient-specific alterations in x-ray attenuation. Further, we utilized a sucrose water phantom due to the lack of standard phantoms for muscle density analysis. While we did not test stability of the phantom, the primary purpose of the phantom was for comparison with internal calibration rather than continued use of a phantom. For this study, we examined muscle density values for the same muscle sample across different scan positions and protocols. However, we did not examine derived muscle density values across different scanners or scan manufacturers, limiting the generalizability of our findings on internal calibration reliability.

## Conclusions

We developed a new internal calibration method to measure muscle density from CT images that does not require calibration phantoms and is robust across scan protocols and positioning within the scanner bore. This internal calibration method will enable the secondary analysis of clinically acquired CT images to assess muscle density, a quantitative assessment of muscle quality, using an inexpensive method without exposure to additional ionizing radiation.

## Supporting information

**S1 Fig. Regression plot comparing calibration methods.** Regression plots comparing the muscle density values derived from the hydroxyapatite (HA) bone phantom method with A) the reference sucrose water phantom method and B) the internal calibration method (n = 10

muscle samples). Solid line indicates regression line, dashed line indicates the line of unity.
(TIFF)

**S2 Fig. Bland-Altman plots comparing calibration methods.** Bland-Altman plots comparing calibration methods. The mean differences are shown as solid lines and 95% limits of agreement are shown as dashed lines. A) Bland-Altman plot comparing differences between internal derived muscle density values and hydroxyapatite (HA) bone phantom derived muscle density values (n = 10 muscle samples). B) Bland-Altman plot comparing differences between the sucrose water phantom derived muscle density values and hydroxyapatite (HA) bone phantom derived muscle density values (n = 10 muscle samples). Internal and sucrose water phantom derived muscle density values are represented in $g/cm^3$. HA phantom derived muscle density values are represented in $mgHA/cm^3$.
(TIFF)

**S1 File. Data used to generate Figs 3–5, S1 and S2, and mean and standard deviation values reported.**
(XLSX)

**S2 File. R markdown document with statistical analysis code and outputs.**
(DOCX)

**S3 File. R script files with statistical analysis code.**
(ZIP)

**S4 File. Internal calibration muscle density analysis code.**
(ZIP)

**S5 File. Image registration code.**
(TXT)

## Author Contributions

**Conceptualization:** Ainsley C. J. Smith, Justin J. Tse, Steven K. Boyd, Sarah L. Manske.

**Data curation:** Ainsley C. J. Smith, Sarah L. Manske.

**Formal analysis:** Ainsley C. J. Smith, Justin J. Tse, Tadiwa H. Waungana, Kirsten N. Bott, Michael T. Kuczynski, Sarah L. Manske.

**Funding acquisition:** Ainsley C. J. Smith, Justin J. Tse, Michael T. Kuczynski, Sarah L. Manske.

**Investigation:** Ainsley C. J. Smith, Justin J. Tse, Sarah L. Manske.

**Methodology:** Ainsley C. J. Smith, Justin J. Tse, Sarah L. Manske.

**Project administration:** Ainsley C. J. Smith, Justin J. Tse, Sarah L. Manske.

**Resources:** Ainsley C. J. Smith, Justin J. Tse, Michael T. Kuczynski, Andrew S. Michalski, Sarah L. Manske.

**Software:** Ainsley C. J. Smith, Tadiwa H. Waungana, Kirsten N. Bott, Michael T. Kuczynski, Andrew S. Michalski, Sarah L. Manske.

**Supervision:** Steven K. Boyd, Sarah L. Manske.

**Validation:** Ainsley C. J. Smith, Justin J. Tse, Sarah L. Manske.

**Visualization:** Ainsley C. J. Smith, Sarah L. Manske.

**Writing – original draft:** Ainsley C. J. Smith, Sarah L. Manske.

**Writing – review & editing:** Ainsley C. J. Smith, Justin J. Tse, Tadiwa H. Waungana, Kirsten N. Bott, Michael T. Kuczynski, Andrew S. Michalski, Steven K. Boyd, Sarah L. Manske.

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
