## [Decision Letter · Decision Letter 0]

20 Sep 2022

PONE-D-22-21739Internal calibration for opportunistic computed tomography muscle density analysisPLOS ONE

Dear Dr. Smith,

Thank you for submitting your manuscript to PLOS ONE. After careful consideration, we feel that it has merit but does not fully meet PLOS ONE’s publication criteria as it currently stands. Therefore, we invite you to submit a revised version of the manuscript that addresses the points raised during the review process.

We look forward to receiving your revised manuscript.

Kind regards,

Yosuke Yamada

Academic Editor

PLOS ONE

Journal Requirements:

2. Please include details of the source of the animals samples in your methods section.

Reviewers' comments:

Reviewer's Responses to Questions

**Comments to the Author**

1. Is the manuscript technically sound, and do the data support the conclusions?

Reviewer #1: Yes

Reviewer #2: Yes

2. Has the statistical analysis been performed appropriately and rigorously? 

Reviewer #1: Yes

Reviewer #2: Yes

3. Have the authors made all data underlying the findings in their manuscript fully available?

Reviewer #1: Yes

Reviewer #2: Yes

4. Is the manuscript presented in an intelligible fashion and written in standard English?

Reviewer #1: Yes

Reviewer #2: Yes

5. Review Comments to the Author

Reviewer #1: The authors designed and validated an internal density calibration method for estimating muscle density from CT images based on established calibration methods in the evaluation of bone mineral density. The findings from this study will contribute to establishing a more accurate method of assessing muscle health. This study is valuable and interesting. I have the following comments.

In this study, bovine bone, bovine adipose, and swine blood were used as ROIs for the internal calibration method. It would be great if you describe the reason more clearly in the Methods Section why you needed to use swine blood.

In this study, the authors used two different scan protocols (abdominal kidney, urine, bladder (KUB) thins protocol and chest mediastinum protocol). Are these protocols widely used?

Page 8, Line 164: “All calibration methods…, and converting the observed HU into muscle density values (g/cm3)”.

Was HA bone phantom included in this “All calibration methods”? The HA bone phantom derives muscle density values in mgHA/cm3 (Page 12, Line 246), so there is a contradiction in the text. I appreciate that you double check it.

Page 10, Line 207: “0.54%”.

What does this percentages mean?

Page 10, Line 207: “We found that with increasing density, the error associated with the internal calibration method diminished linearly (y-intercept = -0.01, slope = 0.006).”

What analysis did the authors use? It would be better if you describe the analysis in the Methods Section. Also, how did you interpret the results obtained? I appreciate that you add it in the Discussion Section.

Are there outliers in the data of this study? (Page 12, Line 240)

How did the authors exclude outliers? It would be better to add the details.

The authors use the word "non-invasive" in the Introduction and Conclusion Sections. However, CT scans are invasive because they involve radiation exposure.

The images in Figures 1 and 2 are unclear. They should be changed to clear images.

Reviewer #2: The authors elaborated an internal density calibration method for assessing muscle density from clinical CT images that is based on a similar method of bone mineral density assessment developed previously. They showed that their method yielded accurate muscle density values when compared with a reference phantom.

In my opinion, the manuscript is well structured and well written. There is ample supporting material provided, including measurement data as well as pieces of code used to perform their statistical evaluations.

I think that the manuscript is of interest for the readership of PLOS One, and I would like to recommend it being considered for publication.

6. PLOS authors have the option to publish the peer review history of their article (what does this mean?). If published, this will include your full peer review and any attached files.

Reviewer #1: No

Reviewer #2: No

---

## [Author Response · Author response to Decision Letter 0]

22 Sep 2022

Response to Reviewers Comments

PONE-D-22-21739

Title: Internal calibration for opportunistic computed tomography muscle density analysis

Submitted to PLOS ONE

We appreciate the thorough review and recommendations provided by the Reviewers. We have responded to each comment and made alterations to the manuscript, shown using track changes, where appropriate.

Reviewer #1: The authors designed and validated an internal density calibration method for estimating muscle density from CT images based on established calibration methods in the evaluation of bone mineral density. The findings from this study will contribute to establishing a more accurate method of assessing muscle health. This study is valuable and interesting. I have the following comments.

In this study, bovine bone, bovine adipose, and swine blood were used as ROIs for the internal calibration method. It would be great if you describe the reason more clearly in the Methods Section why you needed to use swine blood.

We appreciate the reviewers comment regarding the disconnect between species. Given that blood composition is relatively similar between mammals, we prioritized using fresh blood samples, which were more readily available from swine than bovine. We have added the following sentence to explain our reasoning for using swine blood instead of bovine blood (line 107-108), “Swine blood was used instead of bovine blood as it is more readily available as a fresh sample.”

In this study, the authors used two different scan protocols (abdominal kidney, urine, bladder (KUB) thins protocol and chest mediastinum protocol). Are these protocols widely used?

Yes, these scan protocols are regularly used for diagnostic analysis of the abdomen and chest, therefore we utilized these protocols for testing the internal calibration for muscle. The following sentence and references were added (line 118-119), “These scan protocols were selected because they are commonly used in the clinical setting [33,34].”

33. Chen J, Moir D. An estimation of the annual effective dose to the Canadian population from medical CT examinations. J Radiol Prot. 2010;30(2):131–7. doi: 10.1088/0952-4746/30/2/002

34. Berdahl CT, Vermeulen MJ, Larson DB, Schull MJ. Emergency department computed tomography utilization in the United States and Canada. Ann Emerg Med. 2013;62(5). doi: 10.1016/j.annemergmed.2013.02.018

Page 8, Line 164: “All calibration methods…, and converting the observed HU into muscle density values (g/cm3)”.

Was HA bone phantom included in this “All calibration methods”? The HA bone phantom derives muscle density values in mgHA/cm3 (Page 12, Line 246), so there is a contradiction in the text. I appreciate that you double check it.

We appreciate the reviewer raising this point as this sentence also references the HA bone phantom calibration. We have modified the sentence to reflect this point. (line 166-169) “All calibration methods involved manually segmenting the bovine muscle samples in ITK-SNAP V3.8.0 [22], a medical image visualization and analysis software, and converting the observed HU into muscle density values (g/cm3 or mgHA/cm3).”

Page 10, Line 207: “0.54%”.

What does this percentages mean?

We have added the following text to provide context for the reported 0.54%. (line 207-211) “The internal muscle density calibration method slightly underestimated muscle density, but the error was low when compared to the reference sucrose water phantom method (< 0.006 g/cm3) with only a 0.54% difference between the muscle density values derived from the two methods (Fig 3).”

Page 10, Line 207: “We found that with increasing density, the error associated with the internal calibration method diminished linearly (y-intercept = -0.01, slope = 0.006).”

What analysis did the authors use? It would be better if you describe the analysis in the Methods Section. Also, how did you interpret the results obtained? I appreciate that you add it in the Discussion Section.

We have added details for this analysis to the methods and discussion sections. (line 183-184), “Linear regression was performed to assess any association between bias and the magnitude of density.” (line 283-285), “We found that our internal calibration method was more accurate, when compared to the sucrose water phantom, with increasing density values. Hence, this method may be more accurate when applied to denser muscle.”

Are there outliers in the data of this study? (Page 12, Line 240)

How did the authors exclude outliers? It would be better to add the details.

Thank you for catching this oversight. As there were no outliers, we have removed the wording from the Fig. 5 legend.

The authors use the word "non-invasive" in the Introduction and Conclusion Sections. However, CT scans are invasive because they involve radiation exposure.

Thank you for this comment, we have modified our wording accordingly. (line 61-65), “Muscle cross-sectional area and density can be assessed through CT imaging. Chest and abdominal CT images are frequently acquired in hospital settings for clinical diagnoses and these clinically acquired CT scans have been repurposed to investigate muscle cross-sectional area, and muscle attenuation as a surrogate for density, at the level of the lumbar vertebrae without exposure to additional ionizing radiation [14–17].” (line 309-311), “This internal calibration method will enable the secondary analysis of clinically acquired CT images to assess muscle density, a quantitative assessment of muscle quality, using an inexpensive method without exposure to additional ionizing radiation.” 

The images in Figures 1 and 2 are unclear. They should be changed to clear images.

We acknowledge that the image in Figure 1 and 2 do not appear clear in the manuscript preview. During the submission process PLOSONE noted the images may appear unclear in the preview and that the full resolution would be used in the final manuscript for crisper images.

Reviewer #2: The authors elaborated an internal density calibration method for assessing muscle density from clinical CT images that is based on a similar method of bone mineral density assessment developed previously. They showed that their method yielded accurate muscle density values when compared with a reference phantom.

In my opinion, the manuscript is well structured and well written. There is ample supporting material provided, including measurement data as well as pieces of code used to perform their statistical evaluations.

I think that the manuscript is of interest for the readership of PLOS One, and I would like to recommend it being considered for publication.

---

## [Decision Letter · Decision Letter 1]

2 Oct 2022

Internal calibration for opportunistic computed tomography muscle density analysis

PONE-D-22-21739R1

Dear Dr. Smith,

We’re pleased to inform you that your manuscript has been judged scientifically suitable for publication and will be formally accepted for publication once it meets all outstanding technical requirements.

Kind regards,

Yosuke Yamada

Academic Editor

PLOS ONE

Additional Editor Comments (optional):

Nice experiments and idea. Thank you submitting to PLOS ONE.

Reviewers' comments:

Reviewer's Responses to Questions

**Comments to the Author**

1. If the authors have adequately addressed your comments raised in a previous round of review and you feel that this manuscript is now acceptable for publication, you may indicate that here to bypass the “Comments to the Author” section, enter your conflict of interest statement in the “Confidential to Editor” section, and submit your "Accept" recommendation.

Reviewer #1: All comments have been addressed

2. Is the manuscript technically sound, and do the data support the conclusions?

Reviewer #1: Yes

3. Has the statistical analysis been performed appropriately and rigorously? 

Reviewer #1: Yes

4. Have the authors made all data underlying the findings in their manuscript fully available?

Reviewer #1: Yes

5. Is the manuscript presented in an intelligible fashion and written in standard English?

Reviewer #1: Yes

6. Review Comments to the Author

Reviewer #1: (No Response)

7. PLOS authors have the option to publish the peer review history of their article (what does this mean?). If published, this will include your full peer review and any attached files.

Reviewer #1: No

---

## [Editor Report · Acceptance letter]

7 Oct 2022

PONE-D-22-21739R1 

Internal calibration for opportunistic computed tomography muscle density analysis 

Dear Dr. Smith:

I'm pleased to inform you that your manuscript has been deemed suitable for publication in PLOS ONE. Congratulations! Your manuscript is now with our production department. 

Kind regards, 

on behalf of

Dr. Yosuke Yamada 

Academic Editor

PLOS ONE